# Transcatheter Arterial Embolization (TAE) Using N-Butyl-2-cyanoacrylate (NBCA) as the First Choice for Postpartum Vulvovaginal Hematoma; Case Series and Reviews of the Literature

**DOI:** 10.3390/diagnostics12061429

**Published:** 2022-06-09

**Authors:** Takuya Misugi, Akihiro Hamuro, Kohei Kitada, Yasushi Kurihara, Mie Tahara, Eisaku Terayama, Ken Kageyama, Akira Yamamoto, Daisuke Tachibana

**Affiliations:** 1Department of Obstetrics and Gynecology, Graduate School of Medicine, Osaka Metropolitan University, Osaka 545-8585, Japan; tmisugi@omu.ac.jp (T.M.); g21621t@omu.ac.jp (K.K.); v21555o@omu.ac.jp (Y.K.); mtahara@omu.ac.jp (M.T.); dtachibana@omu.ac.jp (D.T.); 2Department of Diagnostic and Interventional Radiology, Graduate School of Medicine, Osaka Metropolitan University, Osaka 545-8585, Japan; b21213l@omu.ac.jp (E.T.); kageyama@omu.ac.jp (K.K.); d21070o@omu.ac.jp (A.Y.)

**Keywords:** transcatheter arterial embolization, N-butyl-2-cyanoacrylate (NBCA), vulvovaginal hematoma, complications

## Abstract

Transcatheter arterial embolization (TAE) has long been reported to be safe, effective, and to have a high clinical and technical success rate for vulvovaginal hematoma. We used a permanent embolic material, diluted N-butyl-2-cyanoacrylate (NBCA), for the first choice intervention for six cases of vulvovaginal hematoma, in order to confirm the effectiveness of NBCA. Regarding post-embolization adverse events, we did not observe any fever nor necrosis or pain in the vaginal wall or vulva, in all cases. The use of NBCA as a first-line treatment for TAE of vulvovaginal hematoma is considered to be effective, in the following two ways: First, hemostasis can be achieved by adjusting the mixing ratio of NBCA and lipiodol, according to the distance between the tip of the catheter and the site of injury. Second, NBCA does not cause complications such as pain, necrosis, or infection, and it can be used safely. There are no reports clearly recommending NBCA as the first choice in the treatment of TAE for vulvovaginal hematoma. This is the first report to examine the efficacy and safety of NBCA as the first-line intervention for such cases.

## 1. Introduction

Postpartum hemorrhage (PPH) remains the leading cause of maternal mortality, and transcatheter arterial embolization (TAE) has long been reported to be safe, effective, and to have a high clinical and technical success rate [1]. According to a report compiled by the Death Case Review and Evaluation Committee (Japan Obstetrics and Gynecology Association) in 2018, the rate of maternal mortality due to PPH accounts for 20% of all maternal mortality and 78 deaths in the past 10 years [2]. The absorbent gelatin sponges used for uterine artery embolization due to uterine bleeding cause a temporary obstruction, because blood vessels reopen in about 3 to 6 weeks after surgery. Therefore, these sponges are the most widely used embolic materials for the purpose of minimizing damage to the uterus [3].

Vulvovaginal wall hematoma is a disease in which blood vessels around the vulvovaginal wall are hyperextended and compressed due to delivery; thus, causing genital tract injury. The hematoma accumulates in the vulva, vaginal mucosa, or coarse connective tissue, and vulvar hematoma is caused by injury of the posterior rectal artery, perineal artery, posterior labial artery, etc., which are branches of the pudendal artery. Vaginal hematoma is often caused by injury of the descending branch of the uterine artery. These blood vessels are small in diameter, and may be occluded proximal to the injured blood vessel by the use of large particle materials such as gelatin sponges. This proximal obstruction may develop collaterals and may also cause further bleeding [4,5]. In uterine artery embolization, permanent embolic materials may cause uterine necrosis and uterine cavity adhesions; therefore, gelatin sponges may be useful as the first choice for embolic materials. However, there have been no reports on an embolic substance in TAE for vulvovaginal hematoma caused by the injury of a blood vessel whose diameter is smaller than that of the uterine artery.

This is the first report to examine the efficacy and safety of N-butyl-2-cyanoacrylate (NBCA) as the first choice for a TAE embolic material for vulvovaginal hematoma. Although it is a preliminary report with only six cases, it is possible that NBCA may be a better choice of embolic substance in such cases.

## 2. Materials and Methods

Six cases using NBCA as the first choice for vulvovaginal hematoma performed at Osaka City University Hospital from August 2017 to April 2021 were included. The inclusion criteria in this study were patients treated with TAE for postpartum vulvovaginal hematoma (with/without uterine hemorrhage). We excluded patients treated with only surgical treatment or expectant management. During that period, 25 cases of vulvovaginal hematoma were observed (included 5 cases at uterine bleeding). Of these, 19 cases improved with surgical treatment and expectant management, and six cases underwent TAE treatment. In all six cases, we used NBCA as the first choice embolic material. Two of the 6 cases were after delivery at our hospital, and four cases were transported from another hospital. The treatment algorithm for vulvovaginal hematoma at our hospital is shown in Figure 1.

We now describe the treatment protocol for PPH at our institution. We judge that PPH has occurred when the Shock Index >1 or 1000 mL or more of bleeding (vaginal delivery) is observed. We then check vital signs and bleeding volume, as well as the bleeding site, by transabdominal ultrasonography and a pelvic examination. Furthermore, we perform a blood examination, infuse adequate fluids, and then consider a blood transfusion. In cases of bleeding from the uterine cavity, we search for the cause, including atonic bleeding, placental remnants, amniotic fluid embolism, etc. If the bleeding persists, we check a contrast-enhanced CT and consider hemostasis, including TAE or hysterectomy. If disseminated intravascular coagulation (DIC) is observed, anti-DIC treatment, as described later, is performed. In cases of bleeding from the vaginal wall, we surgically stop the bleeding. If a vulvovaginal hematoma is found, we proceed with the treatment shown in Figure 1. In DIC management, we evaluate and utilize the following three points, according to the Japanese “obstetrical DIC score”: (1) any underlying disease that might increase the chances of DIC, (2) clinical symptoms (maternal conditions caused by DIC), and (3) laboratory findings (blood examination abnormalities caused by DIC). Based on this obstetrical DIC score, a score of eight points or higher is judged to be a DIC state [6]. From then on, transfusion therapies (FFP transfusion, platelet concentration transfusion, RBC transfusion, albumin transfusion, and fibrinogen products), as well as anti-DIC therapies (antithrombin products, recombinant thrombomodulin), are considered.

In the case of transportation from another hospital, we contact the interventional radiologist and anesthesiologist in our hospital when the transportation request is put through. Upon the patient’s arrival, we first check vital signs and any sign of DIC; we then prioritize blood transfusions and anti-DIC treatment, as needed. If it is possible to make a pelvic examination, an obstetrician will make a diagnosis of vulvovaginal hematoma and confirm the presence of extravasation by contrast-enhanced CT. If there is extravasation, we consult an interventional radiologist and consider TAE treatment. The procedure of TAE by the interventional radiologist at our hospital is shown below. The femoral artery is retrogradely punctured from the right inguinal region, a 4Fr sheath is placed, and the extravasation is confirmed by internal iliac artery angiography using a MOHRI catheter^®^ (Medikit Co. Ltd., Tokyo, Japan). In cases where uterine bleeding is also present, uterine artery embolization is performed at the same time using a gelatin sponge (Spongel^®^, LTL Phama Co., Shinjuku-Ku, Japan). In the case of vulvovaginal hematoma, the complication is often caused by the injury of blood vessels, such as the cervical vaginal branch of the uterine artery, the vaginal branch of the inferior gluteal artery, and the branch of the internal pudendal artery. If the vessel of extravasation is confirmed, we immediately advance the catheter as close as possible to the injured vessel. When the vessel diameter is small, or the vessel is highly flexed, we use a microcatheter Marvel tri-axial system (2.7-Fr microcatheter, Carnelian^®^ HF and 1.9-Fr no-taper microcatheter, MARVEL^®^, Tokai Medical, Kasugai, Japan). When we use NBCA, we dilute it to NBCA:lipiodol = 1:4. In addition, we use it diluted up to 1:12, taking into account the diameter of the vessel and the distance between the tip of the catheter and the injured area. The obstetrician will perform a transvaginal examination at the end of the TAE, determine that there is no tendency for the hematoma to grow, and consult with the radiologist to complete the TAE. Considering the possibility of re-bleeding, we leave the sheath until the next day and remove it if it does not re-bleed. This study was approved by our hospital’s institutional review board (Institutional review board approval number: 2021-095, and the paper conforms to the provisions of the Declaration of Helsinki), and we have been given informed consent in writing from all patients.

## 3. Results

The backgrounds of the six cases are shown in Table 1. The median age was 31 years (range: 27–33 years), median BMI was 22.0 (range: 18.5–26.4), and there were five primiparas and one multiparous woman. The modes of delivery were spontaneous vaginal delivery in two cases, accelerated delivery in two cases, and vacuum extraction in two cases. In addition to vulvovaginal hematoma, uterine bleeding was also present in three of the six cases, so we used a gelatin sponge in cases where uterine artery embolization was necessary. Regarding the actual use of NBCA, we present the procedure for using NBCA in case 5 as an example. As shown in Figure 2a, we found a linear shadow and hematoma on the right side of the vaginal wall in the contrast CT arterial phase, so we consulted with a radiologist and decided on the TAE. The radiologist placed a 4Fr sheath and performed contrast enhancement, which revealed extravasation of the vaginal branch of the right pudendal artery (Figure 2b,c). Although the vessel diameter was small, we were able to advance the microcatheter to the vicinity of the injured vessel. We then embolized with diluted embolic material (NBCA:lipiodol = 1:4) (Figure 2d). We did not find extravasation on the inferior gluteal angiography and then determined that hemostasis was finished (Figure 2e). Cases 1 and 6 were cases in which severe bleeding occurred. Case 1 was a delivery with chronic myelogenous leukemia. We attempted operative hemostasis three times after vaginal delivery but did not succeed. Therefore, TAE was performed the day after delivery and hemostasis was possible. Case 6 also had atonic bleeding with DIC (obstetrical DIC score 18 point). Case 3 was a case in which bleeding occurred from the collateral tract, due to a proximal embolism (Figure 3). We placed a 4Fr sheath and found extravasation from the vaginal branch of the left uterine artery. We tried to advance the microcatheter, but it could not advance to the vicinity of the bleeding point, due to strong bending and meandering. Therefore, we embolized the vaginal branch with diluted embolic material proximally (NBCA:lipiodol = 1:4) (Figure 3a–d). We then confirmed new bleeding from the vaginal branch of the left inferior gluteal artery by internal iliac artery angiography. Again, we were unable to advance the microcatheter due to severe flexion and meandering, so we switched to the Microcatheter Marvel tri-axial system^®^ and were able to advance the catheter to the vaginal branch. We then embolized the vaginal branch with diluted embolic material (NBCA:lipiodol = 1:12), to prevent proximal embolization, and were finally able to stop bleeding (Figure 3e,f). Regarding post-embolization adverse events, we did not observe any fever, nor necrosis or pain, in the vaginal wall or vulva in all cases. Subsequent pregnancy has not been confirmed in any of the six cases.

## 4. Discussion

In the 1970s, embolization for life-threatening bleeding was published for the upper abdominal gastrointestinal tract [7], cases of pelvic fracture [8], and malignancies [9], and Brown et al. reported on uterine artery embolization in 1979 [10]. Regarding obstetric bleeding, Paull et al. reported on a balloon occlusion of the abdominal aorta in 1995 [11], Dubois on a balloon occlusion and cesarean section of the internal iliac artery in 1997 [12], Weeks et al. on internal iliac artery balloon occlusion and cesarean section in 2000 [13], and Shih et al. published a temporary occlusion of the common iliac artery and cesarean section in 2005 [14]. In addition, many other reports have reported on the benefits of IVR for bleeding. The embolic substances of TAE are mainly gelatin sponges, coils, and NBCA. It is common to embolize with a piece of gelatin sponge of appropriate size, usually 1 to 2 mm, and the size of gelatin sponge fragments varies depending on the radiologists’ experience, so that the expected outcome may be different [15,16,17].

NBCA is used in combination with lipiodol, and the time to embolization can be adjusted to some extent by the mixing ratio. In an in vitro experiment, Stoesslein et al. reported the time to embolic materials in blood as follows: NBCA:lipiodol = 1:1 for 3.2 ± 0.8 s, 1:2 for 4.7 ± 0.5 s, 1:3 for 7.5 ± 1.5 s, and 1:4 for 11.8 ± 1.5 s [18]. Since the mixing ratio can be adjusted for each case, and occlusion in the proximal region can be avoided, it is considered to be effective for TAE of vulvovaginal hematoma with small blood vessel diameters. On the other hand, this report also pointed out that inexperienced people should avoid using it alone, because assessing the distance between the tip of the catheter and the site of injury of the blood vessel takes some getting used to.

In a study of embolic material in TAE treatment for PPH, Tanahashi et al. reported a discussion on conversion from gelatin sponge to NBCA. The discussion indicated the height of the uterine cavity and systolic blood pressure as risk factors for conversion [19]. Furthermore, in the report, there are reports of uterine necrosis and adhesions of the uterine lumen regarding the use of NBCA for the uterus, and it is stated that the use of NBCA requires caution [20].

As the number of cases have increased, factors behind obstetric TAE failure have also been reported. The causes of TAE failure are listed as follows: (1) history of uterine scars and past arterial ligation that interferes with catheter placement, (2) occurrence of uterine artery spasm, (3) embolization of a single artery or proximal embolization causes collateral circulation, and (4) DIC condition [15]. As in (3) listed here, it has been mentioned that proximal occlusion may cause further bleeding, due to the development of collaterals. We had a similar experience in Case 3 of this study. For small-diameter vascular structures such as the vulvovaginal vessels, gelatin sponges tend to cause proximal occlusion; and NBCA, which can adjust the time to embolization, is useful for embolizing smaller blood vessels.

We searched the keywords “vulvovaginal hematoma”, “embolization”, and “genital tract injury (trauma)” in PubMed [5,19,21,22,23,24,25,26,27,28]. Among them, we found papers that describe embolic materials (Table 2). In the reviews, fever and pain were reported as minor complications; however, there are many cases in which uterine artery embolism was also performed, and it was indistinguishable as a complication of TAE for vulvovaginal hematoma. Only one case was found having wound dehiscence of episiotomy in the review. No complications such as fever, pain, or wound dehiscence were observed in the six cases we report in this study.

Therefore, we think that one of the possibilities for the low number of complications in TAE of vulvovaginal hematoma is that the amount of NBCA used is extremely low. This is because the vessel is often thinned and injured at one point. As shown in Table 1, all cases used less than 1 mL of NBCA, which is extremely low compared to the TAE of other organs.

Although there have been reports of cases using NBCA for vulvovaginal hematoma, there are no reports on the use of NBCA as a first-line embolic material or the adjustment of NBCA concentration vis-à-vis the position of the catheter and the injured blood vessel.

Furthermore, there are reports that TAE using NBCA is effective for cases with DIC condition. Obata et al. reported the usefulness of NBCA when hemostasis was difficult due to abnormal coagulation [29,30]. In our report, hemostasis was possible, even in a case complicated with DIC, as presented in Case 6.

## 5. Conclusions

The use of NBCA as a first-line treatment for TAE of vulvovaginal hematoma is considered to be effective in the following two ways: First, hemostasis can be achieved by adjusting the mixing ratio of NBCA and lipiodol, according to the distance between the tip of the catheter and the site of injury. Second, NBCA does not cause complications such as pain, necrosis, or infection, and since no such reports have been recognized, it can be used safely. However, as a limitation, there are only six cases in this study, and it is, therefore, necessary to accumulate more cases.

## Figures and Tables

**Figure 1 diagnostics-12-01429-f001:**
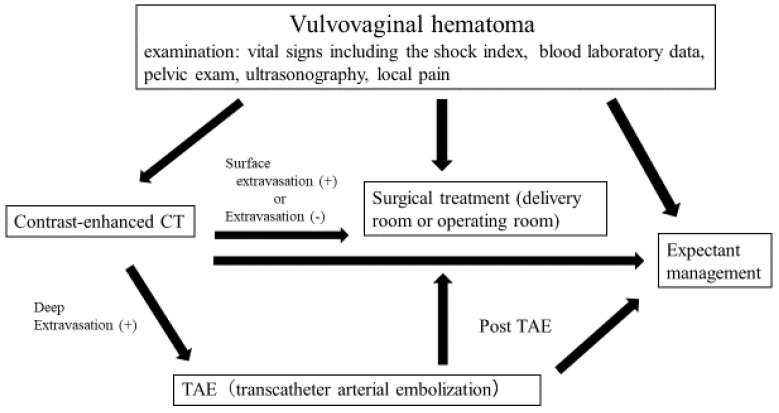
Algorithm for vulvovaginal hematoma treatment at our hospital.

**Figure 2 diagnostics-12-01429-f002:**
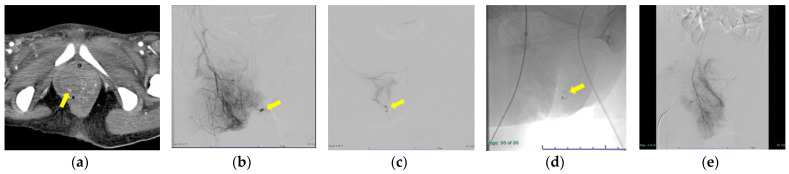
The procedure for using NBCA (in Case 5). (**a**) Contrast enhanced-CT arterial phase. (Arrow) A linear shadow and hematoma were found on the right side of the vaginal wall, and arterial bleeding was suspected. (**b**) Right pudendal arteriography. (Arrow) Extravasation was observed in the vaginal branch. (**c**) (Arrow) Extravasation was observed from the peripheral branch of the internal pudendal artery. (**d**) Since the catheter could be advanced to the vicinity of the bleeding point, embolization was performed with NBCA diluted solution (NBCA:lipiodol = 1:4). The arrow is an image of NBCA accumulating at the bleeding point. (**e**) Inferior gluteal angiography. Disappearance of extravasation image.

**Figure 3 diagnostics-12-01429-f003:**
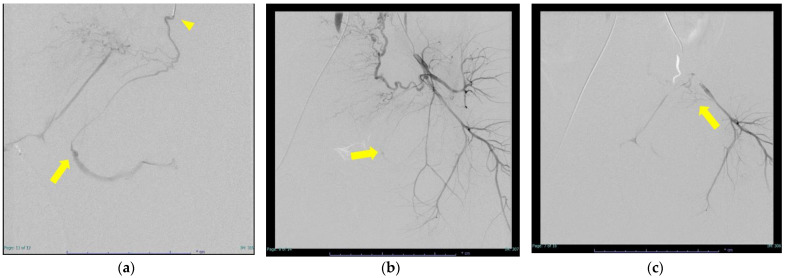
Collateral tract due to a proximal embolism (in Case 3). (**a**) Vaginal branch of uterine artery angiography. (Arrow) Extravasation is observed. (Arrowhead) The catheter could not be advanced any further, and embolization was performed with NBCA diluted solution (NBCA:lipiodol = 1:4). NBCA did not reach the bleeding point, resulting in a proximal embolism. (**b**) Post-embolization internal iliac artery angiography. (Arrow) Bleeding images are still observed via the collateral tract. (**c**) (Arrow) Bleeding remained from the small diameter collateral tract branching from the inferior gluteal artery. (**d**) (Arrow) The catheter could not be advanced to the vicinity of the bleeding point. (**e**) Since the blood vessels were very fine, embolization was performed using an NBCA diluted solution (NBCA:lipiodol = 1:12). (**f**) Left internal iliac artery angiography. Extravasation image has disappeared.

**Table 1 diagnostics-12-01429-t001:** Characteristics of patients.

Case	Age	BMI	Weeks at Delivery (W/D)	Primiparity /Multiparity	Mode of Delivery	Hospital of Delivery	Weight of Birth (g)	Total Blood Loss (mL)	Shock Index	Plt (×10^3^/μL)	PT-INR	FBG (mg/dL)	Transfusion	Underlying Disease	Operative Hemostasis before InterventiOnal Radiology	Extravasation (Embolic materials) and Amount of Diluted NBCA
1	28	18.5	31/0	primiparity	induction of labor	own	1426	6300	1.06	64.7	1.13	171	RBC20U, FFP22U, cryoprecipitate8U	Chronic myeloid leukemia	operative hemostasis	right internal pudendal artery branch (NBCA:lipodol = 1:4) 0.3 mL bilateral uterine artery (gelatin sponge)
2	30	25	41/4	primiparity	vacuum extraction	other	3340	1385	1.07	26	0.96	505	―		―	two vaginal branches of the lower left buttock artery (NBCA:lipiodol = 1:4 & NBCA:lipiodol = 1:3) 0.6 mL left uterine artery (gelatin sponge)
3	22	26.4	39/6	primiparity	vacuum extraction	other	3052	2050	0.91	21	1.02	271	RBC12U, FFP8U	Hypertensive disorders of pregnancy	―	uterine artery vaginal branch (NBCA:lipodol = 1:4) 0.2 mL bleeding from posterior collateral tract due to proximal obstruction 0.2 mL vaginal branch from the lower left buttock artery (NBCA:lipodol = 1:12) 0.3 mL
4	30	23.2	40/4	primiparity	normal vaginal delivery	other	2986	1670	1	15.1	0.88	338	RBC8U		―	internal pudendal artery branch (direct branch from superior gluteal artery bifurcation level) (NBCA:lipodol = 1:3) 2 times 0.5 mL
5	27	23.3	37/5	primiparity	normal vaginal delivery	other	2472	2100	0.54	14.5	0.96	228	RBC6U, FFP4U	Hypertensive disorders of pregnancy	operative hemostasis	right internal pudendal artery branch (NBCA:lipodol = 1:4) 0.4 mL
6	33	21.8	40/2	multiparity	induction of labor	own	2530	12700	2.4	3	2.32	37	RBC36U, FFP30U, PC80U cryoprecipitate12U	Gestational diabetes mellitus Cervical cerclage	―	left internal pudendal artery branch (NBCA:lipodol = 1:6) 0.3 mL left uterine artery (gelatin sponge) left obturator artery (NBCA:lipodol = 1:4) 0.4 mL

**Table 2 diagnostics-12-01429-t002:** Reviews of vulvovaginal hematoma treated by TAE [5,19,21,22,23,24,25,26,27,28].

Author	Year of Publication	Journal Title	Cases	Year	Nulliparity/ Multiparity	Total Blood Loss (mL)	Complications	Extravasation	Embolic Materials
Linda J.	1985	American Journal of Perinatology	3	34	Nulliparity	1500	none	right and left pudendal arterys	gelatin sponge
				27	Multiparity	800	none	brench of the left hypogastric artery	gelatin sponge
				27	Multiparity	1500	none	left internal pudendal artery	gelatin sponge
Homer G.	1989	Am J Obstet Gynecol	2	36	Multiparity	4000	fever	vaginal branch of the internal pudendal artery	gelatin sponge
				24	Multiparity	3000	fever	branch of internal iliac artery	gelatin sponge
Y. Yamashita	1990	Amerian College of Obstetrican and Gynecologists	6	no description	no description	no description	fever (one case)	anterior division of internal illiac artery (4 case) left internal illiac artery (one case) left internal pudendal artery (one case)	gelatin sponge
J. Villella	2001	J Reprod Med	2	32	Multiparity	2000	none	distal branch of the internal pudendal artery	gelatin sponge
				31	Nulliparity	no description	none	no description	gelatin sponge
Elias M.D.	2013	Case Rep Obstet Gynecol	1	29	Nulliparity	750	situs inversus totalis	distal branch of the anterior part of the right lower iliac artery	gelatin sponge
A. Takeda	2014	European Journal of Obstetrics & Gynecology and Reproductive Biology	4 (*1)	20–39	no description	50–4000	no description	left vaginal artery	gelatin sponge → coil
								vaginal branch of the left uterine artery	gelatin sponge
								descending branch of the internal iliac artery	gelatin sponge
								branch of the left uterine artery	gelatin sponge
K. Takagi	2017	Taiwan J Obstet Gynecol	2	32	Nulliparity	729	none	branch of the right internal pudendal artery	gelatin sponge
				34	Nulliparity	821	none	left internal pudendal artery	gelatin sponge
Sang Min Lee	2018	European Radiology	60	31.5 (28–40)	Nulliparity: 52 Muitiparity: 8	no description	pneumonia and left ventricular dysfunction: 1 fever: 7 oedema of the lower legs: 2 pancreatitis: 2 irregular menstruation: 2	vaginal artery: 24 uterine artery (cervicovaginal branch): 18 internal pudendal artery: 13 cervical artery: 9 inferior mesenteric artery: 4 external pudendal artery: 3 obturator artery: 2 inferior vesical arterya: 1	gelatin sponge particles: 23 gelatin sponge with permanent embolic agents (microcoils, NBCA): 34 permanent embolic agents only: 3
Swati Shivhare	2021	Turkish Journal of Obstetrics and Gynecology	2	28	Nulliparity	no description	none	brunch of internal pudendal artery	gelatin sponge
				26	Multiparity	no description	none	right vaginal artery pseudoaneurysm	NBCA (*2)
Hyun Jung Lee	2021	J Vasc Interv Radiol	43	32.6 ± 4.63	Nulliparity: 33 Muitiparity: 10	no description	early (<7 d after procedure) pulmonary edema: 11 (25.6%) febrile morbidity: 10 (23.3%) other complications (wound dehiscence): 1 (2.3%) none: 24 (55.8%) late (>8 d after procedure) amenorrhea or oligomenorrhea: 2 (4.7%) pain: 4 (9.3%)	vaginal artery: 9 (20%) internal pudendal artery: 5 (11.1%) round ligament or ovary artery: 6 (13.3%) uterine artery: 19 (42.2%) cervicovaginal branch: 4 (8.9%) internal iliac artery: 1 (2.2%) vesical artery: 1 (2.2%)	gelatin sponge: 9 (23.1%) permanent agents: 34 (76.9%) microcoils, histoacryl glue
K. Sasaki	2021	Emergency Radiology	27	31.8 ± 5.7	Nulliparity: 17 Muitiparity: 10	no description	no major complications	vaginal artery;17 perineal artery: 8 internal pudendal artery: 4 obturator artery: 3 inferior mesenteric artery: 1 inferior rectal artery: 1	gelatin sponge: 12 NBCA or coils: 15

(*1) Excerpt only TAE case for vulvovaginal hematoma. (*2) NBCA was performed for TAE for pseudoaneurysm.

## Data Availability

Not applicable.

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
