# Peer review of "Transcatheter Arterial Embolization (TAE) Using N-Butyl-2-cyanoacrylate (NBCA) as the First Choice for Postpartum Vulvovaginal Hematoma; Case Series and Reviews of the Literature"

_diagnostics, 2022, doi:10.3390/diagnostics12061429_

Round 1

Reviewer 1 Report

The authors demonstrated 6 cases of vulvovaginal hematoma treated by TAE with NBCA. Although the efficacy of TAE with NBCA has been established in the postpartum hemorrhage, the reports regarding vulvovaginal hematoma are limited so this report has a clinical value.

  • The author may want to clarify the inclusion and exclusion criteria of this study. All of the patients with vulvovaginal hematoma were treated with NBCA during this period? No patients were treated with TAE with gelatin sponge (GS)? If there are some treated with GS, why they are not treated with NBCA?
  • Related to the above question, are there any criteria for which embolization material (i.e, NBCA or GS) will be used in the authors' hospital?
  • The way of describing the statistical data "The age was 31 years (median: 93 27-33 years), BMI was 22.0 (median: 18.5-26.4)" is not appropriate. This should be "The median age was 31 (range: ....)"
  • Figure: space between "P" "elvic" should be modified. Also, in the same square, more explanation would be needed (e.g., Examination: vital signs....). Also, the term "Blood check" is odd. Does this mean blood test/laboratory data? 
  • Informed consent waived?

Author Response

Thank you very much for providing important comments. We are thankful for the time and energy you expended.

Our responses to the referees’ comments (Reviewer 1) are as follow:

  1. The author may want to clarify the inclusion and exclusion criteria of this study.

We have added the following sentence in the material and methods:

“The inclusion criteria in this study are patients treated with TAE for postpartum vulvovaginal hematoma (with/without uterine hemorrhage). We excluded the patients treated with TAE for only uterine hemorrhage.”

・All of the patients with vulvovaginal hematoma were treated with NBCA during this period?

Yes. Vulvovaginal hematoma sometimes has active bleeding. As well, some patients fall into a DIC state. Therefore, we need to stop bleeding as soon as possible. That's an additional reason why we use NBCA as the first choice.

・No patients were treated with TAE with gelatin sponge (GS)? If there are some treated with GS, why they are not treated with NBCA?

No, we did not use GS for all vulvovaginal hematomas.

  1. Related to the above question, are there any criteria for which embolization material (i.e, NBCA or GS) will be used in the authors' hospital?

We basically use NBCA for vulvovaginal hematoma as the first choice to achieve secure hemostasis in peripheral vessels, where GS can not reach because of it’s diameter.

  1. The way of describing the statistical data "The age was 31 years (median: 93 27-33 years), BMI was 22.0 (median: 18.5-26.4)" is not appropriate. This should be "The median age was 31 (range: ....)"

      We have modified it as instructed.

  1. Figure: space between "P" "elvic" should be modified. Also, in the same square, more explanation would be needed (e.g., Examination: vital signs....). Also, the term "Blood check" is odd. Does this mean blood test/laboratory data? 

       We have modified it as instructed.

  1. Informed consent waived?

We have been given informed consent in writing from all patients and with the approval of the Institutional Review Board.

Thank you very much for your important comments.

Yours sincerely,

Akihiro Hamuro

Reviewer 2 Report

This Japanese case series presents six cases of postpartum vulvovaginal hematoma treated with transcatether arterial embolization. The authors present six cases and state that this approach can be used also in cases of established DIC. I have the following comments:

  1. In the introduction section the authors should add the information regarding the maternal mortality in Japan and also how many mothers die from PPH in Japan?
  2. What are the possible approaches for treatment of vulvovaginal hematoma post partum? Is it only interventional radiology? Is contrast enhanced CT always performed or only in cases of circulatory instability? Could it only be managed expectantly with or without vaginal tamponade?
  3. In the materials and methods sections the authors should provide a detailed description of PPH protocol in their institution. Do they perform transabdominal ultrasound in case of uterine bleeding to exclude retained placental parts or just proceed to interventional radiology in all cases? A detailed DIC management protocol should also be described.
  4. Are there any international guidelines regarding the management of vulvovaginal hematoma post partum?
  5. I would suggest the authors to omit the sentence that the effect of embolization can be expected in cases of DIC.

Author Response

Thank you very much for providing important comments. We are thankful for the time and energy you expended.

Our responses to the referees’ comments (Reviewer 2) are as follow:

  1. In the introduction section the authors should add the information regarding the maternal mortality in Japan and also how many mothers die from PPH in Japan?

       We have added the following sentence in the introduction:

“According to a report compiled by the Death Case Review and Evaluation Committee (Japan Obstetrics and Gynecology Association) in 2018, the rate of maternal mortality due to PPH accounts for 20% of all maternal mortality and 78 deaths in the past 10 years.”

And then, we added a reference as No.2

  1. What are the possible approaches for treatment of vulvovaginal hematoma postpartum? Is it 

only interventional radiology?

       We will perform surgical treatment or interventional radiology at our institution. However, as a report of other hemostatic methods, there is a report of the usefulness of a vaginal balloon tamponade. However, we have no experience with this procedure at our institution.

・Is contrast enhanced CT always performed or only in cases of circulatory instability?

・Could it only be managed expectantly with or without vaginal tamponade?

    ã€€Contrast-enhanced CT is taken in all cases with vulvovaginal hematoma. At our institution, we do not conservatively manage the patients with a vaginal tamponade. However, as shown in Fig. 1, in some cases where there is no extravasation, no change of hematoma size, no pain, and vital signs are stable, we manage conservatively.

  1. In the materials and methods sections, the authors should provide a detailed description of PPH protocol in their institution.

    We have added the following paragraph in the materials and methods:

    “We now describe the treatment protocol for PPH at our institution. We judge that PPH has occurred when the Shock Index > 1 or 1000 ml or more of bleeding (vaginal delivery) is observed. We then check vital signs and bleeding volume, as well as the bleeding site, by transabdominal ultrasonography and a pelvic examination. Furthermore, we perform a blood examination, infuse adequate fluids, and then consider a blood transfusion. In cases of bleeding from the uterine cavity, we search for the cause, including atonic bleeding, placental remnants, amniotic fluid embolism, etc. If the bleeding persists, we check a contrast-enhanced CT and consider hemostasis, including TAE or hysterectomy. If DIC is observed, anti-DIC treatment, as described later, is performed. In cases of bleeding from the vaginal wall, we surgically stop the bleeding. If a vulvovaginal hematoma is found, we proceed with the treatment shown in Fig.1. In DIC management, we evaluate and utilize the following three points and according to the Japanese “obstetrical DIC score”: 1) any underlying disease that might increase the chances of DIC, 2) clinical symptoms (maternal conditions caused by DIC), and 3) laboratory findings (blood examination abnormalities caused by DIC). Based on this obstetrical DIC score, a score of 8 points or higher is judged to be a DIC state. From then on, transfusion therapies (FFP transfusion, platelet concentration transfusion, RBC transfusion, albumin transfusion, and fibrinogen products), as well as anti-DIC therapies (antithrombin products, recombinant thrombomodulin), are considered.”

     And then, we added a reference as No.6

・Do they perform transabdominal ultrasound in case of uterine bleeding to exclude retained placental parts or just proceed to interventional radiology in all cases?

   If bleeding persists with PPH, we perform contrast-enhanced CT to first check for bleeding points. We also perform transabdominal ultrasound as described in the Materials and Methods section.

・A detailed DIC management protocol should also be described.

   We added our DIC management protocol in the Materials and Methods section.

  1. Are there any international guidelines regarding the management of vulvovaginal hematoma postpartum?

   There are many reports of surgery and IVR, but there are no international treatment guidelines for vulvovaginal hematoma.

  1. I would suggest the authors to omit the sentence that the effect of embolization can be expected in cases of DIC.

   We have omitted the sentence that the effect of embolization can be expected in cases of DIC.

Thank you very much for your important comments.

Yours sincerely,

Akihiro Hamuro

Reviewer 3 Report

This is a very well written series of cases presenting usefulness of N-butyl-2-cy- 2 anoacrylate (NBCA) as a clotting substance in transcatheter artery embolization procedure applied in obstetric patients with postpartum lower genital tract injury.

The authors suggest to apply NBCA as first line management in vaginal and vulvar hematomas based on fewer side effects compared to so far standard approach with absorbent gelatin sponges, effectiveness in DIC and better possibility to close smaller vessels than in other techniques.

To improve excellent manuscript I suggest to add expansions to all abbreviations on tab.1 e.g. IVR, CML, HDP.

In tab.1 correction is needed for "accelated vaginal delivery" in column 6.

It would be interesting to compare costs and other characteristics of presented NBCA technique with alternative so far standard hemostatic procedures since it is proposed as first choice postpartum therapy.

Author Response

Thank you very much for providing important comments. We are thankful for the time and energy you expended.

Our responses to the referees’ comments (Reviewer 3) are as follow:

・To improve excellent manuscript I suggest to add expansions to all abbreviations on tab.1 e.g. IVR, CML, HDP.

We have modified it as instructed.

・In tab.1 correction is needed for "accelated vaginal delivery" in column 6.

We have modified it as instructed.

・It would be interesting to compare costs and other characteristics of presented NBCA technique with alternative so far standard hemostatic procedures since it is proposed as first choice postpartum therapy.

From the cost aspect, NBCA costs 4800 yen for a vial and gelatin sponge costs 9040 yen for a sheet in Japanese market. However, it depends on how many vials and sheets are to be used. We added following sentences to compare the characteristics in the first paragraph of discussion; “and the size of gelatin sponge fragments varies depending on the radiologists’ experience so that the expected outcome might be different.”

Thank you very much for your important comments.

Yours sincerely,

Akihiro Hamuro

Round 2

Reviewer 1 Report

Thank you for addressing the comments.

1. “The inclusion criteria in this study are patients treated with TAE for postpartum vulvovaginal hematoma (with/without uterine hemorrhage). We excluded the patients treated with TAE for only uterine hemorrhage.

> The authors may want to add the number of the patients who met the initial inclusion criteria and how many were excluded, because when the inclusion and exclusion criteria are shown, the number of the patients should be demonstrated. 

2. Informed consent waived? We have been given informed consent in writing from all patients and with the approval of the Institutional Review Board.

> Please add this to the manuscript. 

Reviewer 2 Report

All comments have been addressed and I believe the manuscript is now suitable for publication

Author Response

Dear Editor and Reviewer

Thank you very much for providing important comments. We are thankful for the time and energy you expended.

Yours sincerely,

Akihiro Hamuro